# Photocatalytic CO_2_ Conversion into Solar Fuels Using Carbon-Based Materials—A Review

**DOI:** 10.3390/molecules28145383

**Published:** 2023-07-13

**Authors:** Dhivya Sundar, Cheng-Hua Liu, Sambandam Anandan, Jerry J. Wu

**Affiliations:** 1Department of Environmental Engineering and Science, Feng Chia University, Taichung 407, Taiwan; dhivyasundar9896@gmail.com (D.S.); chhengliu@fcu.edu.tw (C.-H.L.); 2Department of Chemistry, National Institute of Technology, Trichy 620015, India; sanand99@yahoo.com

**Keywords:** carbon materials, photocatalytic CO_2_ reduction, derived chemical fuels, Z-scheme heterojunction

## Abstract

Carbon materials with elusive 0D, 1D, 2D, and 3D nanostructures and high surface area provide certain emerging applications in electrocatalytic and photocatalytic CO_2_ utilization. Since carbon possesses high electrical conductivity, it expels the photogenerated electrons from the catalytic surface and can tune the photocatalytic activity in the visible-light region. However, the photocatalytic efficiency of pristine carbon is comparatively low due to the high recombination of photogenerated carriers. Thus, supporting carbon materials, such as graphene, CNTs (Carbon nanotubes), g-C_3_N_4_, MWCNs (Multiwall carbon nanotubes), conducting polymers, and its other simpler forms like activated carbon, nanofibers, nanosheets, and nanoparticles, are usually combined with other metal and non-metal nanocomposites to increase the CO_2_ absorption and conversion. In addition, carbon-based materials with transition metals and organometallic complexes are also commonly used as photocatalysts for CO_2_ reduction. This review focuses on developing efficient carbon-based nanomaterials for the photoconversion of CO_2_ into solar fuels. It is concluded that MWCNs are one of the most used materials as supporting materials for CO_2_ reduction. Due to the multi-layered morphology, multiple reflections will occur within the layers, thus enhancing light harvesting. In particular, stacked nanostructured hollow sphere morphologies can also help the metal doping from corroding.

## 1. Introduction

Global climate change due to the rise in global warming and the energy crisis due to the depletion of fossil fuels have become significant threats to the Earth in recent years. The rising human population, excessive transportation use, and rapid industrialization have all contributed to the excessive usage of fossil fuels, which has led to increased CO_2_ emissions into the atmosphere. The combustion of fossil fuels, such as petrol, oil, diesel, natural gas, and biogas, is the major source of CO_2_ emission. Though several gases contribute to environmental pollution, CO_2_ emission is the utmost among them. The dependence on fossil fuels has grown significantly, particularly since the industrial revolution, which will prevent the next few decades’ worth of coal, oil, and natural gas reserves from being fully utilized.

Reports on greenhouse gas emissions and their detrimental effects on the atmosphere’s natural balance are frequently released. Figure 1a depicts a new assessment by the International Energy Agency (IEA) on the global extreme CO_2_ emissions from industrial processes and energy combustion from 1900 to 2021. The report states that the amount of gas being released into the environment has increased annually and will reach 36.3 billion tons in 2021. As was already mentioned, CO_2_ addresses both the question of global warming and the issue of energy. As a result, it is constantly imperative and incumbent upon researchers to address the problems related to CO_2_ emission. The capture, storage, and conversion of carbonic acid gas to its derivatives are the three fundamental steps for the research community to reduce the rate of CO_2_ release into the environment and utilize its derivatives for further energy needs. There are numerous processes available for the reduction of CO_2_ into its derivatives, such as photochemical, electrochemical, thermochemical, and photo-biochemical approaches with certain both benefits and shortcomings. Thermochemical processes demand high temperatures, while electrochemical processes have high voltaic requirements. Biological activities additionally necessitate ambient temperature and pressure. Among them, photochemical or photocatalytic techniques may become the most suitable technique for photocatalytic redox reactions [1]. It avails natural resources, such as water and sunlight for CO_2_ reduction. Semiconductor materials with appropriate bandgap and efficient catalytic properties are frequently used as stand-alone catalysts and composites with variously doped metal oxides, especially TiO_2_ metal sulfides, perovskites oxides/halides, and graphitic carbon nitrides (g-C_3_N_4_), as shown in Figure 1b [2].

Carbon is one of the Earth’s most abundant, widely distributed elements. Carbon-based materials have been largely used in photocatalytic CO_2_ reduction. Considering their vast availability, low cost, and eco-friendliness, carbon-based materials with elusive 0D, 1D, 2D, and 3D nanostructures and high surface area provide certain emerging applications in electrocatalytic and photocatalytic CO_2_ utilization, solar cells, batteries, supercapacitors, biosensors, and energy storage and conversions [4]. Graphene [5], graphene oxides [6], carbon nanotubes (CNTs) [7], g-C_3_N_4_ [8], conducting polymers [9], and carbon aerogels [10] are the most used carbonaceous materials for photocatalysis. These are the most prevailing catalysts, exhibiting a better catalytic efficiency both as a stand-alone and supporting material [11]. Carbon materials are excellent supporters as they could highly help in improving the adsorption and activation of CO_2_. In addition, they also play a major role in a wide range of light absorption and help in enhancing the stability of the catalysts with which it has been composited. 

This review elucidates the advantages of utilizing carbon materials for photocatalytic CO_2_ reduction. In the front part of the review, we make a brief note on the fundamental ideas underlying photocatalytic CO_2_ reduction and the basic requirements that should be satisfied by a photocatalyst to efficiently participate in CO_2_ reduction. In the later part, we make a detailed discussion about the contribution of carbon materials in enhancing the efficiency of the catalysts using some of the recently published works.

## 2. Principles of Photocatalytic CO_2_ Reduction

The most promising method for reducing the high levels of greenhouse gases in the atmosphere involves photocatalytic processes since they can provide the required thermal energy for their effective reduction into their fuel derivatives [12]. However, there are still challenges associated with each step of its photocatalytic reduction. Here are the four principal steps that have been involved in CO_2_ reduction into its derivatives: adsorption of light energy by the photocatalyst, generation of electron–hole pairs, transportation of the charge carriers, and the chemical reaction between the charge carriers and surface species [13]. 

The first and foremost task in photocatalysis is to activate a semiconductor by absorbing a maximum amount of natural sunlight or from artificial light energy sources. The photons of light energy that are being absorbed should be equal to or more than those of the bandgap energy of the semiconductor photocatalysts [14]. After absorption of the required light energy, electrons will be generated. The photogenerated electrons become excited and transfer from the completely filled valence band (VB) to the empty conduction band of the semiconductor. The received charge carriers in the conduction band (CB) undergo a redox reaction with the adsorbed surface species on the semiconductor to obtain the aiming reduced products. To be effective, an ideal photocatalyst must have a conduction band edge potential above the reduction potential for the given product, while the valence band edge potential should be well below the oxidation level of the universal solvent, as shown in Figure 2a [15]. Depending on the number of electrons involved, a wide range of C_1_ and C_2_ products, such as methane (CH_4_), methanol, formic acid, and carbon monoxide (CO), can be generated from the proton-assisted multi-electron reduction of CO_2_. CO and CH_4_ are the most obtained products so far [16]. Along with the useful redox reactions of CO_2_, several other backward reactions can also occur. At times, photocatalytic reduction processes become more complex when intermediates and products are converted back into CO_2_. Despite extensive research in this area, the fast electron–hole recombination rate prevents this photocatalytic reduction of CO_2_ from achieving the desired photocatalyst activity and stability [17]. Incorporating carbon materials as supporting materials could help in diminishing these shortcomings and increase photocatalytic efficiency by domineering the forward CO_2_ reduction reaction to solar fuels.

## 3. Multi-Dimensional Carbon-Based Materials for Photocatalytic CO_2_ Reduction

Carbon, a tetravalent non-metallic element that is abundant in the atmosphere, can be classified into three categories based on their dimensionality, such as zero-dimensional (0D) fullerenes, one-dimensional (1D) carbon nanotubes or carbon nano-fibers, two-dimensional (2D) graphite and graphene nanosheets, and three-dimensional (3D) hybrid carbon networks and mesoporous carbons, as shown in Figure 2b. Carbon can be chemically embellished or functionalized using metallic nanoparticles (NPs). The large surface area with defective sites, high porosity, excellent electron conductivity, and chemical inertness makes carbon both a standalone catalyst or catalyst support with semiconductors to obtain an effective reduction of CO_2_. While considering utilizing a standalone catalyst with exceptional physical and chemical stability, low cost, and relatively simple synthesis processes, g-C_3_N_4_ with a layered structure and a moderate bandgap of 2.7 eV is the most popular visible-light active photocatalyst. Most crucially, g-C_3_N_4_ is capable of conducting photocatalytic CO_2_ reduction, which is thanks to its strongly negative conduction band potential of −1.1 V against the standard hydrogen electrode (NHE). As opposed to photocatalysts that incorporate with precious metal and metal salts to prepare, g-C_3_N_4_ is independent of metals and so can be easily made by thermally polycondensing cost-effective N-rich precursors, such melamine, cyanamide, dicyanamide, and urea [18,19]. The carbon-supported catalysts use high-surface-area activated carbon and carbon black as their primary ingredients since they are inexpensive and widely available [20,21]. One-dimensional fullerene (C_60_) is an allotrope that has a highly conjugated three-dimensional structure. The delocalized conjugated structure of such a singular kind gives C_60_ an exceptional electronic property. Particularly, its ability to accept electrons is crucial for electron-transfer processes, as they can render a quick charge separation and a slow carrier recombination [22]. CuO–C_60_ [23] and C_60_ polymer [24] are some of the few examples by which we could observe these heterostructures exhibiting improved charge separation and photocatalytic activity. Nanoscale carbon compounds with at least one dimension smaller than 10 nm are known as carbon dots (CDs), which consist of hybridized carbon atoms with sp^2^ and sp^3^ and have different surface functional groups. The inherent quantity of edges and flaws in carbon dots allows for quantum-confinement processes that boost the exposure of such pyridinic N atoms. As reported in recent work on CDs/CdS [25] and in VCQDs/C_3_N_4_ heterojunctions with carbon vacancy [26], due to the quantum confinement effect, the bandgap is widened in graphene quantum dots in a size-dependent manner. In addition to optical sensing, light-emitting diodes, these 0D graphene carbon dots are also promising catalysts for photocatalysis and energy storage [27]. 

Carbon nanotubes (CNTs) are hollow and one-dimensional materials with nano-capillaries having the capacity to absorb gas molecules, a significant specific surface area of about 1000 m^2^/g, resilient electrical conductivity, and thermal conductivity [28]. These are ideal substrates for heterogeneous catalysis since they are distinctive templates for the immobilization or deposition of metallic nanoparticles [29]. Single-walled carbon nanotubes (SWCNTs) and multi-walled carbon nanotubes (MWCNTs) are the two classes that broadly fall under CNTs. In theory, rolling single-layered graphene in a particular direction to form a cylindrical shape produces a single-walled carbon nanotube [30]. Noble metals, such as Au, Ag, and Pd, are the most used metal catalysts [31], because they could give CO and formate as a reduction product of CO_2_ and also shows a higher selectivity toward it. Considering the availability and cost, non-noble metals, like Cu, Sn, Bi, Pb, In, and their compounds, are usually used as an alternative to the high-cost noble metals [32]. Due to high availability, low toxicity, and less cost, Sn metal is believed to satisfy the needs, and some of the reports on SnO_2_ nanowires and nanosheets show an increase in catalytic active sites and therefore an improved catalytic activity [33]. Therefore, a composition of Sn with carbon supports, like MWCNTs, will create a better conduit for CO_2_ gas diffusion. Cu-MWCNTs/pCN [34] and g-C_3_N_4_-MWCNTs [35] are a few examples that show how MWCNTs act as support for the photocatalysts to obtain better results.

A semiconductor with the finest catalytic activity must have properties including (1) a larger specific surface area with extensive active sites for the potential absorption of absorbate on the catalytic surface; (2) a nanoscale-range thickness or indeed even a reduced range to sub-nanometer scale for an effective charge-transfer process; and (3) a 2D flexible planar structure with the possibility of making modifications, like heterojunction formation and vacancy introduction. [36]. Carbon-based materials with π–π conjugation can encourage the flow of charge carriers and provide a large surface area for efficient dispersion. The three types of 2D materials that are most frequently utilized in photocatalysis are transition metal dichalcogenides, graphene, and g-C_3_N_4_ [37]. Hexagonally packed 2D graphene nanosheets with π–π have exceptional thermal and electrical conductivity and abundant CO_2_ adsorption. With a zero-band-gap energy, it could absorb almost the full arc light spectrum. Due to its one-atom-thick structure, it is usually regarded as having the highest specific surface area of any material [38]. Graphene materials with 0D, 2D, 3D, and a combination of all of the three-dimensional graphene-based catalysts have been widely used for photocatalytic CO_2_ reduction [6,39]. g-C_3_N_4_ is an n-type semiconductor and a member of the wider group of materials known as covalent organic polymers, which also include conjugated porous polymers [40,41], covalent organic frameworks [42,43], and covalent triazine frameworks [44,45]. g-C_3_N_4_ has a moderate band gap between 2.7 and 2.8 eV, where it allows for direct photocatalytic CO_2_ conversion because of its sufficiently negative conduction band (CB) edge that may convert CO_2_ into valuable hydrocarbon [46]. Other than graphene and g-C_3_N_4_, in recent days, graphdiyne, a novel 2D carbon material, has begun to make progress. It is made up of aromatic rings comprising sp- and sp^2^-carbon as well as carbon–carbon triple bonds connected by acetylenic linkages. In order to reach the most stable state, electrons in the high-energy acetylenic links are prone to delocalization, and as a result, they are likely to end up acting as the active centers for adsorption and potential catalysis [47]. The photocatalytic performance of porous 3D porous carbon could possess a greater photocatalytic activity due to its wide surface area, multiple reactive sites, and porous shape. Carbon–carbon networks with a larger surface area along with the porous structure on a nanometer scale highly help CO_2_ reduction to obtain C_1_ and C_2_ products.

## 4. Advantages of Utilizing Carbon Materials for CO_2_ Reduction

The advantages of using carbon materials for photocatalytic reactions include (1) enlarging the specific surface area, (2) improving CO_2_ adsorption and activation, (3) separating photogenerated charges, and (4) increasing light absorption.

### 4.1. Enlarging the Specific Surface Area

Chemical and structural modifications on the surfaces of solid materials possessing high surface areas have been investigated to improve CO_2_ adsorption and selectivity [48]. Carbon-based materials, such as graphene, graphene oxides, carbon nanotubes (CNTs), and g-C_3_N_4_, provide greater surface area to the catalysts where it is composited [49]. In the case of an increase in surface area, there will be an increase in the adsorption of CO_2_ molecules. Carbon compounds also function as capping agents to control the growth of nanoparticles, increasing the specific surface area of the catalysts. [50]. g-C_3_N_4_ has a high proportion of surface defects since it contains a number of hydrogen atoms and possesses electron-rich properties. It has proven helpful in catalysis due to its high value in encouraging electron relocalization on the catalytic surface. Yang et al. designed NiAl-layered double hydroxide nanosheets (NALDH) and then combined them with g-C_3_N_4_ nanosheets, and witnessed an ultra-tight nanosheet–nanosheet heterojunction. They also added graphene aerogels, which further helped in expanding the structure to a network framework. In this work, they utilized both graphene nanosheets and aerogels to increase the efficiency of the photocatalyst. The formation of ultra-tight sheet–sheet heterojunctions helped in shortening the distance for charge transfer, and it also provided abundant active sites. Graphene aerogel with a 3D structure can act as excellent catalytic support and provide a larger surface area for CO_2_ adsorption. The exceptional absorptivity and stability of the catalyst resulted in the production of 4 and 16 times more pure NALDH and bare carbon nitride. A wide range of increased specific surface areas and pore volume was calculated using N_2_ adsorption–desorption isotherms, as shown in Figure 3a [51]. CNTs with peculiar structures and a nanometric diameter range have a larger specific surface area [52]. Nitrogen-doped porous carbon nanofibers were recently coupled with a dispersion of nickel and molybdenum phosphide (Mo/Ni-PS@PAN) by Chen et al. They followed a phosphatizing strategy to increase the size of the catalyst and achieved an increased diameter of 20 nm to 50 nm. Carbonization was performed to further increase the size. The combination of nanoparticles of MoP with highly dispersed Ni atoms resulted in a NiMoP@NC_PF_ material. As a result of carbonization, the size of the catalyst increased from 50 nm to 100 nm, as shown in Figure 3b,c. The specific surface area of Ni-MoP@NC_PF_ was 60.5 m^2^ g^−1^, which was larger than that of Ni-MoP@NC_SF_ (30.7 m^2^ g^−1^), showing that NiMoP@NC_PF_ has a much higher amount of mesopores than NiMoP@NC_SF_. The obtained porous structure of the material helped in promoting the CO_2_ adsorption for the photoreduction reaction as shown in Figure 3d [53].

### 4.2. Increasing the CO_2_ Adsorption and Activation

In addition to the pore structures, the nature of the gas–surface interaction is also crucial for CO_2_ adsorption. Thus, surface-modified carbons could polarize CO_2_ molecules and thereby improve their adsorption by introducing basic functional groups into the carbon framework [54]. There have been several porous adsorbents investigated for their potential to capture CO_2_, such as zeolites, metal–organic frameworks (MOFs), porous carbons, and organic–inorganic hybrid sorbents [55]. Among them, carbon materials with several forms have been used as supporting materials for CO_2_ photoreduction. Materials based on carbon have great potential as highly efficient materials. The tunable morphology and high surface area make it an efficient adsorbent for CO_2_ [56]. As we have already seen, g-C_3_N_4_ itself could possess a greater surface area. In some cases, graphene materials can be protonated using protonating agents to increase their specific surface area. In recent work, Wu et al. made a novel work in which they have g-C_3_N_4_ as a template and fabricated layers of g-C_3_N_4_ along with a co-doping of Ni/Co metals in the cavities present in g-C_3_N_4_. g-C_3_N_4_ material was protonated using phosphoric acid. With the addition of a varied proportion of metal dopings, there was a change in microstructure and differently sized g-C_3_N_4_-Co_1.6_Ni_0.4_ were obtained, as shown in Figure 4a–c. The addition of bimetallic doping created numerous holes in g-C_3_N_4_ and thus increased the nitrogen vacancies. Individual g-C_3_N_4_ itself showed a CO production rate of about (3.49 mmol g^− 1^ h^− 1^), but the addition of metal doping enhanced the CO_2_ adsorption by creating nitrogen vacancies. Figure 4d depicts the catalytic activity of the catalyst at different proportions. The photocatalytic CO_2_ reduction of g-C_3_N_4_ and g-CN-Co_x_Ni_y_ resulted in CO as the primary product, demonstrating the strong selectivity of CO. This illustrates the advantage of the extended p-p conjugated system and the red shift in the visible edge in the bimetal-doped g-C_3_N_4_. Most importantly, the porous structure can offer a lot of sites and a wide surface area for photocatalytic activity, which accelerates the reaction. Thus, this helped with the enhanced CO production rate of 13.5l mmol g^− 1^ h^− 1^, which was 3.9 times higher than that of g-C_3_N_4_, as shown in Figure 4e [57].

### 4.3. Separation of Photogenerated Electron–Hole Pairs

Carbon materials have good conductive properties. The photogenerated electron–hole transfer is the second step in photocatalytic CO_2_ reduction, which takes place after the absorption of light by the catalyst. When light of a variety of wavelengths strikes the catalysts, the electrons in the valence band are excited and migrate to the conduction band. Along the path of the transfer, charge recombination may occur during the electron transfer process, which reduces the number of electrons accessible to participate in the forward redox reaction. To prevent the recombination of charges, hole scavengers are therefore required. The highest capacity for capturing and retaining electrons is found in carbon-based materials, which makes them interesting to be considered [58].

Being the most widely used supporting semiconductor material, g-C_3_N_4_ with a larger specific surface area has drawn a lot of attention due to its exceptional merits in expanding the range of visible-light absorption, improving photoexcited charge separation, and offering more active sites for CO_2_ activation, but it suffers from a major disadvantage of fast charge recombination of charges. A viable photocatalyst should have a larger bandgap that correlates with a higher oxidation/reduction ability. At the same time, a reduced bandgap boosts the light-harvesting capacity and will raise the efficiency of solar energy utilization simultaneously. In order to enhance the efficiency of the photocatalytic system, it is crucial to design a heterojunction between these kinds of single catalysts to help in overcoming these challenges. The creation of heterojunction devices has the potential to be successful due to the separation of photogenerated electron–hole pairs and the combination of the distinctive efficiencies of both catalysts [59]. The charge transfer pathway in heterostructure photocatalysts typically follows either a type II heterojunction or a Z-scheme diagram depending on the arrangement of the band-positions and Fermi-levels of the heterojunction components [60]. Liu et al. used a morphology-inherited strategy to design a hollow-scheme SnS_2_/g-C_3_N_4_/C (SCC) photocatalyst. The formation of a Z-scheme heterojunction highly helped in increasing the CO_2_ adsorption sites and thus increased the production rate. It also increased the reduction ability of electrons in the CB edge of g-C_3_N_4_. The introduction of hollow carbon spheres improved the quantity of light energy utilized, increased the dispersion of SnS_2_/g-C_3_N_4_ for more reactive active sites, and also the capacity of photogenerated electrons transfer. Figure 5a shows the electron transport mechanism of SCC, in which the obtained Z-scheme photocatalysts follows a typical type II heterojunction mechanism and the electrons formed on g-C_3_N_4_ will be transferred to the CB of SnS_2_. Since the calculated conduction band potential of SnS_2_ (−0.46 eV) in this work is higher than the theoretical CO_2_-CO reduction band potential (−0.53 eV), additional energy is needed to push the redox reaction of CO_2_ to CO. The Z-scheme formation greatly aids electron–hole separation. Electrons on the CB of SnS_2_ are speedily transmitted to the VB of g-C_3_N_4_ through the conductive carbon sphere, where they are combined with the photogenerated holes. The internal electric field holds back the hole transfer on the VB of SnS_2_. Therefore, the Z-scheme system greatly helps to separate the electron and holes and enhance the charge flow for CO_2_ reduction to CO [61]. 

The increase in the separation of charges decides the lifetime and stability of the material. Wang et al. fabricated a layered g-C_3_N_4_/rGO/NiAl-LDHs heterojunction. Reduced graphene oxide (rGO) was used as an electron transfer bridge between the g-C_3_N_4_ and NiAl-LDHs and facilitated a rapid charge migration. During the electron transfer mechanism, electrons became excited to both CB edges of Ni_3_Al-LDHs and g-C_3_N_4_. The interfacial internal electric field at the interface and the type II heterojunction facilitated the photogenerated electron transfer from the VB of the g-C_3_N_4_ to the VB of the Ni_3_Al-LDHs. The photogenerated holes accelerated in the CB of g-C_3_N_4_, which advanced the electron–hole charge separation, as shown in Figure 5b. The usage of an excellent conductive rGO material resulted in an increased CO production of 14.2 times that of pure g-C_3_N_4_ [62].

### 4.4. Increasing Light Absorption

Increased visible-light photocatalytic activity is brought about by the inclusion of carbon materials like MWCNT, graphene, fullerenes, carbon nanoparticles, and g-C_3_N_4_ [63]. As we know already, TiO_2_ is the most used metal oxide catalyst for photocatalysis of its appropriate bandwidth to absorb visible-light photocatalysis [64]. Camarillo et al. used two supercritical methods to prepare CNT/TiO_2_ and CNT/TiO_2_/Cu nanocomposites. The addition of CNT greatly aided the synthesized composites to exhibit a greater catalytic activity than bare TiO_2_. The majority of CNT/TiO_2_ composites have been created through either mechanically combining CNT and TiO_2_ or through the use of a coating, sol–gel, and solvothermal techniques. Here, in this work, they used superfluids for preparing the composite. In addition to facilitating the elimination of ligand breakdown products by desorption during decompression, this encourages the transfer of the precursor to the fluid/solid interface at high concentrations. Mere TiO_2_ was mentioned as P25, different ratios of CNT and TiOs were taken, and in addition to that, they doped Cu metal over CNT/TiO_2_ composite to find out more about the catalytic efficiency with the addition of metal doping.

As depicted in Figure 6, the introduction of CNT made a great impact on narrowing the bandgap of the prepared composites and enhanced the absorption in the visible-light region. The charge transfer mechanism to the nanotubes took place effectively since there was less radiative recombination of photoinduced electrons trapped on the TiO_2_ particle surface. Even without the presence of metal doping, CNT50/TiO_2_50 exhibited the best photocatalytic performance among the prepared composites by showing a wide range of absorbance in the visible-light region. While considering CNT/TiO_2_/Cu ternary composites, the effect of metal doping was shielded by CNT. Therefore, an optimal ratio of carbon supporters should be used to obtain maximum productivity [65]. 

Surface modification by defect engineering and nanostructuring are some of the strategies that have been followed in recent years to construct a defect-rich photocatalyst surface. Yang et al. created an accordion-like structure in a nitrogen-vacancy defect, including C-doped graphitic carbon–nitride nanosheets (g-CN-X) for the first time utilizing the steam technique and ethanol. The enhanced g-CN-10 demonstrated an improved visible-light response over the whole visible-light spectrum. The designing of a defect-rich catalytic surface leads to a more localized charge density distribution and promotes photocatalytic active sites, which increase the light absorption efficiency and improve the transport of charge carriers involved in photocatalysis. A graphical model of the location of nitrogen-vacancy defects over g-CN-X with C-doping is shown in Figure 7a. Due to the gaps between the ultrathin nanosheets in the accordion structure, there was the formation of more mesopores and macropores of g-CN-X, which were higher than those of bulk g-CN, as depicted in Figure 7b. Hence, the accordion-like morphology with a distributed hierarchy surface effectively promotes photoconversion efficiency. The energy gap of the g-CN-X (1.82–2.48 eV) was precisely controlled, resulting in an improved light absorption throughout the visible spectrum region, as shown in Figure 7c,d [66]. The efficiency of photo-energy conversion in a non-metal carbon-based nanomaterial can be enhanced by designing a defect-rich catalytic surface, which helps in increasing visible-light absorption for the photocatalytic reduction reaction.

## 5. Conclusions

As seen in Table 1, a wide range of typical carbon materials, like CNT, g-C_3_N_4_, conducting polymers, and its other simpler forms like activated carbon, nanofibers, nanosheets, and nanoparticles, have been used with other metal and non-metal nanocomposites for photocatalytic CO_2_ reduction. These materials with varied dimensions and morphology were used as supporters with photocatalysts for photocatalytic CO_2_ reduction to increase the selectivity, stability, and reducing ability. Carbonaceous materials could highly help in improving the photocatalytic CO_2_ reduction mechanism in each of the stages. These are good electron acceptors for aiding in rapid electron transfers. Graphene and rGO materials possess higher surface area, which helps in the adsorption of an adequate quantity of CO_2._ Since it has a larger surface area to wrap around the photocatalyst, it could avail a uniform distribution of nanoparticles and helps in reducing the size of the nanoparticles. In addition, it is concluded that MWCNs are one of the most used materials as supporting materials for CO_2_ reduction due to their multi-layered morphology and stacked nanostructured hollow sphere morphologies, which can help the metal doping from corroding.

Carbon materials, like graphene, can absorb almost the whole spectrum of full arc light. If the amount of light absorbed is high, then there will be more production and supply of photogenerated electrons to the redox mechanism. Another advantage is that carbon materials could be used as photosensitizers to increase visible-light absorption. Despite the numerous advantages that carbon materials can offer for photocatalysts to increase efficiency, there are still some disadvantages that exist for practical application, where the photocatalyst’s reusability is a crucial parameter that needs to be evaluated. At times, the addition of rGOs substantially decreases the stability of the photocatalysts. To improve the stability, heterostructured CNTs along with nitrogen vacancies could be used to further increase the stability of the catalysts. Since carbon materials are black in color, it absorbs the maximum wavelength of light, which in turn creates a negative photothermal effect, called the shielding effect. An excessive addition of carbon materials could shield the efficiency of metal dopants on the catalytic surface. These fascinating features of these carbon-based materials can be used and evaluated to construct highly selective and more effective photocatalysts for CO_2_ reduction application.

## Figures and Tables

**Figure 1 molecules-28-05383-f001:**
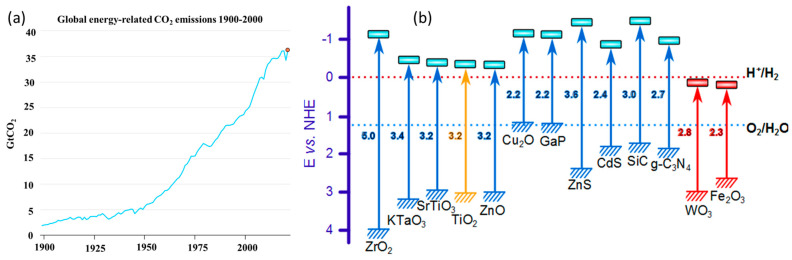
(**a**) A graph on CO_2_ emissions from energy combustion and industrial processes, 1900–2021. IEA report 2021. (**b**) Semiconductors with tunable bandgap for CO_2_ reduction. Reproduced with permission from reference [3]. Copyright 2019 Elsevier.

**Figure 2 molecules-28-05383-f002:**
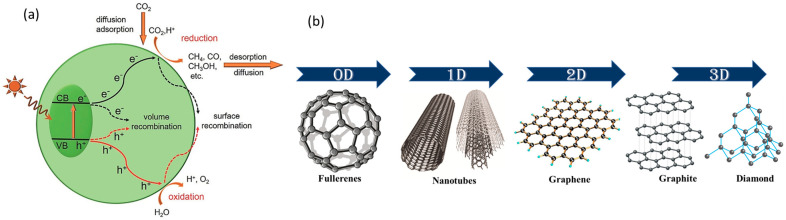
(**a**) Mechanism of photocatalytic CO_2_ reduction [15]. (**b**) Carbon materials classification into three categories based on their dimensionality.

**Figure 3 molecules-28-05383-f003:**
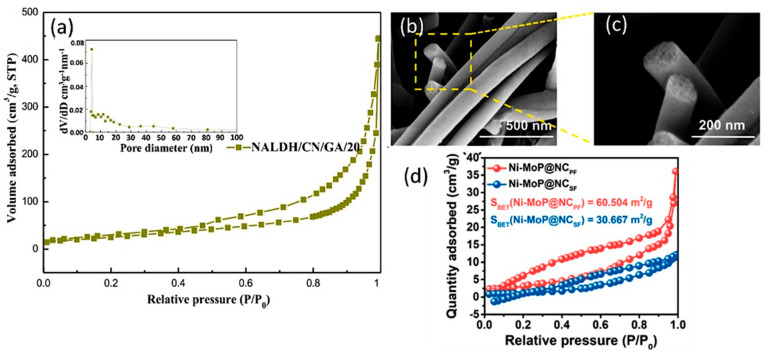
(**a**) N_2_ adsorption–desorption isotherms and the corresponding pore size distribution images (the inset) of NALDH. Reproduced with permission from reference [51]. Copyright 2022 Elsevier. (**b**,**c**) SEM images of Ni-MoP@NCPF. (**d**) BET data of Ni-MoP@NCPF and Ni-MoP@NCSF. Reproduced with permission from reference [53]. Copyright 2022 Elsevier.

**Figure 4 molecules-28-05383-f004:**
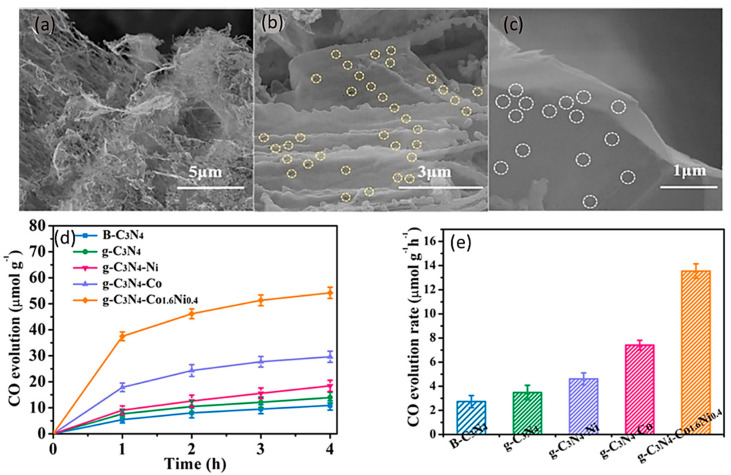
(**a**–**c**) SEM images of g-C_3_N_4_-Co_1.6_Ni_0.4_ in different sizes, (**d**) CO evolution over time, (**e**) CO evolution rate by g-C_3_N_4_-Co_x_Ni_y._ Reproduced with permission from reference [57] Copyright 2022 Elsevier.

**Figure 5 molecules-28-05383-f005:**
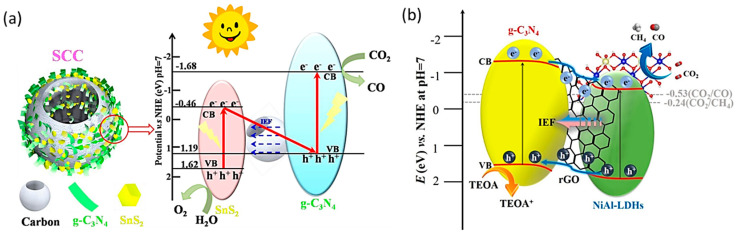
(**a**) Mechanism on photocatalytic reduction of CO_2_ into CO on the SCC. Reproduced with permission from reference [61]. Copyright 2022 Elsevier. (**b**) Charge transfer mechanism of g-C_3_N_4_/rGO/Ni_3_Al-LDHs composite. Reproduced with permission from reference [62]. Copyright 2022 Elsevier.

**Figure 6 molecules-28-05383-f006:**
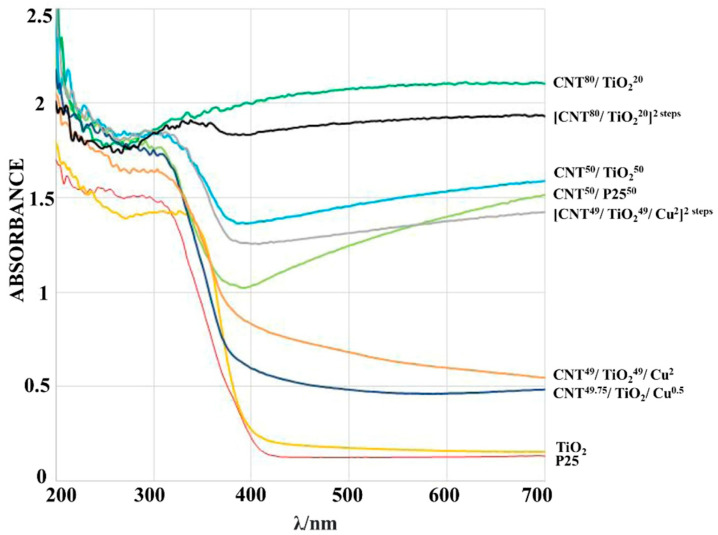
UV-vis DRS spectra of P25 and TiO_2_ catalysts produced in supercritical media, both in their pure form and as composites. Reproduced with permission from reference [65]. Copyright 2020 Elsevier.

**Figure 7 molecules-28-05383-f007:**
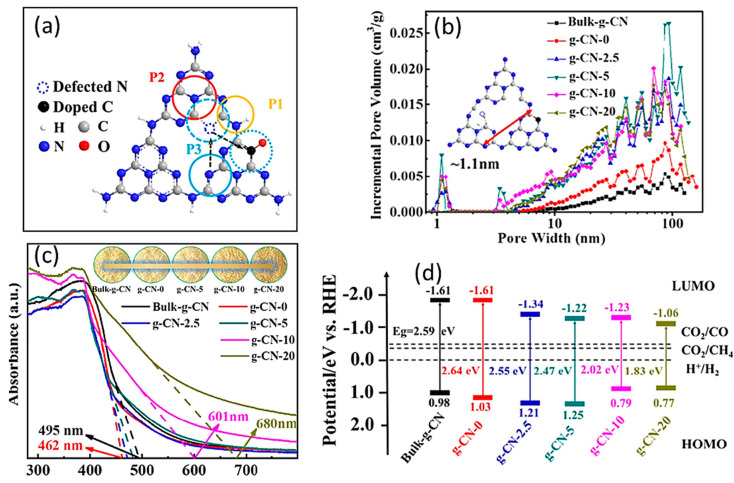
(**a**) Distributions of pore sizes in bulk-g-CN and g-CN-X. (**b**) A graphical model of N-vacancies defects over C-doped g-CN-X, (**c**) UV–vis DRS, and (**d**) bandgap structure for different g-CN-X samples. Reproduced with permission from reference [66]. Copyright 2022 Elsevier.

**Table 1 molecules-28-05383-t001:** A wide variety of typical carbon materials as supporters for photocatalytic CO_2_ reduction.

S. No	Carbon Materials as Supporters	Advantages	Findings	Ref. No
1	g-CNQDs@MOF	Increased surface area	The addition of g-CNQDs enhanced the specific surface area of the binary composite from 338.6 m^2^/g to 828.3 m^2^/g	[67]
2	NiO/g–C_3_N_4_–carbon microsphere composites	Increased surface area	g-C_3_N_4_ and carbon microsphere were added as templates for a larger surface area.	[68]
3	Band-gap-engineered g-C_3_N_4_/rGO	charge separation and conductivity	The copolymerization technique highly helped for reducing the bandgap. The formation of heterostructure aided in increasing the charge separation	[69]
4	Ni-Ce/eg-C_3_N_4_	Promoted CO_2_ adsorption	eg-C_3_N_4_ created abundant mesopores in the catalytic surface for CO_2_ adsorption	[70]
5	PtCu-crCN	Widespread metal loading	Carbon nitride provided the ligand for the dispersion of isolated Pt and Cu single atoms	[71]
6	N-doped TiO_2_/CNT and N-doped TiO_2_/rGO	Increased the charge recombination time	The addition of rGO doubled the CO_2_ conversion	[72]
7	MXene-based heterojunction photocatalysts	electron/hole reservoirs provided more active sites for CO_2_ adsorption	An optimum amount of Ti_3_C_2_ was used to avoid the stacking effect	[73]
8	C/CdS@ZnIn_2_S_4_ heterojunction photocatalysts	Increased light harvesting	A multi-layer reflection occurred within the stacked structured hollow nanostructure and preserved the metal CdS from corrosion	[74]
9	g-C_3_N_4_/TiO_2_/Ti_3_AlC_2_ 2D/0D/2D composite	Enhanced selectivity	The 2D Ti_3_AlC_2_ MAX acted as a conductive substrate for the 2D/0D S-scheme heterojunction of g-C_3_N_4,_ g-C_3_N_4_ increased the selectivity toward CO_2_ to CH_4_	[75]
10	Z-scheme FeV_2_O_4_/g-C_3_N_4_	Increased charge separation and active sites	The ion exchange liquid chemical.The method combined with the post-annealing technique helped in activity and selectivity	[76]

## Data Availability

Data can be available upon request from the authors.

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
