# Peer review of "Photocatalytic CO2 Conversion into Solar Fuels Using Carbon-Based Materials—A Review"

_molecules, 2023, doi:10.3390/molecules28145383_

Round 1

Reviewer 1 Report

This review paper presented aspects regarding photocatalytic CO2 conversion into solar fuels using carbon-based materials. There are many articles and reviews in specialized journals, but this review addresses to the photocatalytic conversion of CO2 using carbon-based materials.

The review contains the following chapters: introduction, principles of photocatalytic CO2 reduction, multi-dimensional carbon-based materials for photocatalytic CO2 reduction, advantages of utilizing carbon materials for CO2 reduction, and conclusions.

Carbon-based materials have been largely used in photocatalytic CO2 reduction, they having a vast availability, a low cost, eco-friendliness, and high surface area. These properties recommended them to certain emerging applications in electrocatalytic and photocatalytic CO2 utilization, solar cells, batteries, supercapacitors, bio-sensors energy storage and conversions.

The review paper is sound, original, interesting, well-organized and clearly presented.

The introduction could be improved.

You can also take into account the following paper: https://doi.org/10.1016/j.cej.2021.131401

The references are up to date and in accordance with the subject approached (take care to authors names citation!).

I consider that the review paper is of interest and it can be considered for publication after revision.

Please make the following correction:

Page 13, line 427 - 15. Kandy, M.M. Carbon-based photocatalysts for enhanced photocatalytic reduction of CO2 to solar fuels. Sustain. Energy Fuels. 2020, 4, 469-484.

Page 15, line 485 – 37. Low, J,; Yu, J.; Ho, W. Graphene-based photocatalysts for CO2 reduction to solar fuel. J. Phys. Chem. Lett. 2015, 6, 21, 4244-4251.

Author Response

Thank you for your suggestions. The introduction part has been moderately modified in the revised manuscript. In addition, we have cited the reference [1] as suggested and added a few points from the reference in the revised manuscript. We have also checked and corrected the spelling mistake in the reference no. 15 and 37.

Reviewer 2 Report

The article discusses the use of carbon-based materials for photocatalytic CO2 conversion to solar fuels. Carbon materials, such as graphene and carbon nanotubes, have been used as supporters for CO2 reduction, as they enhance CO2 adsorption and activation. Additionally, non-metal and metal composites have been used to increase stability and selectivity, but excessive addition of carbon materials leads to a negative photothermal effect. Carbon materials can be effectively used as catalysts or supporters for photocatalytic CO2 reduction as they possess unique properties such as large specific surface area, π-π conjugation, multiple reactive sites, and porous shape. Carbon-supported catalysts exhibit higher photocatalytic efficiency by dominating the forward CO2 reduction to solar fuels. The author should explain g-C3N4 itself is a good photocatalyst for CO2 reduction which is different from other carbon materials. This paper can be accepted after a mini revision.

Author Response

Thank you for your suggestions. We have added a few more details about g-C3N4 in line 123-131 (page 4) in the revised manuscript.

Reviewer 3 Report

In their work, Wu and coworkers highlighted critically the main points and concerns about the photocatalytic conversion of carbon dioxide by using carbon-based material. I overall think that this work would be highly beneficial for the community and thus I do recommend for publication in molecules upon some revisions.

Introduction,

-       Line 30-31: statements unclear, please reformulate

-       line 32: what do the authors mean by “heavy transportations”?

-       The quality of the images in fig.1a) and 1b) is not really good.

-       “CO2” is repeated 10 times in 14 lines: please reformulate from line 30 to line 44. 

-       “CO2” is repeated 6 times in 5 lines: please reformulate from line 77 to line 82, which looks also redundant.

-        

-       Some other refs are required for the statement ended in line 47.

Chapter 3: I think the statement: “Carbon, a tetravalent non-metallic element that is abundant in the atmosphere, is the primary component of carbon materials” is not really necessary.

Chapter 4.1: there is a problem with the sentence in line 185-186. Please reformulate

Chapter 4.3:  the beginning of this part is a bit unclear. Please reformulate sentences from 247 to 254. Line 268: I think it is worthy to explain what is a “Z-scheme photocatalysis”, also with the support of other references

Extensive english editing is required. Some sentences are complicated or simply unproperly written.

Author Response

Thank you for your kind suggestions.

  1. We have reformulated the line 31-32
  2. We have changed the statement as “excessive transportation use” to avoid any confusing.
  3. Now we have divided Fig. 1(a),(b),(c),(d) into Fig. 1(a)(b) and Fig. 2(a)(b) by enhancing the image quality as seen in the revised manuscript.
  4. We have now reformulated the sentences in such a way as to reduce the repetition of the word “CO2in the revised manuscript.
  5. Chapter 3- We have revised the sentence accordingly.
  6. Chapter 4.1- We have again reformulated the sentences in line 199-201 in the revised manuscript.
  7. Chapter 4.3- We have reformulated the sentences in line 263-270 in the revised manuscript. In addition, the Z-scheme photocatalysts are supported by ref [62].
  8. All English usage has been checked thoroughly in the revised manuscript.